

# Correlation analysis of lipid accumulation index, triglyceride-glucose index and H-type hypertension and coronary artery disease

Wenwen Yuan[1], Yan Shao[1], Dong Zhao[2] and Bin Zhang[1]

[1] Department of Cardiology, Qingdao Fuwai Cardiovascular Hospital, Qingdao, China
[2] Department of ICU, Qingdao Fuwai Cardiovascular Hospital, Qingdao, China

## ABSTRACT

**Objective:** The current research was designed to explore the relationship between the lipid accumulation index (LAP), coronary artery disease (CAD), and the triglyceride-glucose (TyG) index in patient with H-type hypertension.

**Methods:** From June 2021 to January 2022, our hospital's information management system collected data on 186 patients with essential hypertension. The participants were categorized into two groups (H-type hypertension ($n$ = 113) and non-H-type hypertension ($n$ = 73)) based on their homocysteine levels. Both groups' general condition, lipid accumulation index, triglyceride-glucose index, and Gensini score were compared to determine the factors influencing the severity of CAD in H-type hypertension patients.

**Results:** There were statistically significant differences ($P < 0.05$) in homocysteine (Hcy, GLP-1 and SAA) level, LAP, and TyG indexes, but not in body mass index (BMI), smoking, sex, age, total cholesterol (TC), triglyceride (TG), high-density lipoprotein cholesterol (HDL-C), fasting plasma glucose (FPG), diastolic blood pressure, and systolic blood pressure. Additionally, there were substantial variations between the two groups regarding the number of lesion branches, degree of stenosis, and Gensini score ($P > 0.05$). patient with grade III to IV lesions had substantially higher LAP and TyG indices than those with stage I to II ($P < 0.05$). TyG (OR = 2.687) and TyG-LAP (OR = 4.512) were the factors determining the incidence of coronary artery disease in H-type hypertension, according to multivariate logistic regression analysis. The lesion number, stenosis degree, and Gensini score ($P < 0.05$) varied among both groups. LAP and TyG indexes were substantially greater in patients with double and triple vessel lesions than in those without lesions or with single vessel lesions ($P < 0.05$); similarly, these two indexes were considerably higher in individuals with grade III to IV lesions than in patients with grade I to II lesions ($P < 0.05$). As per the Pearson correlation analysis, the LAP, TyG indices and SAAlevel were adversely connected to the Gensini score ($r$ = 0.254, 0.262, 0.299, $P < 0.05$), the GLP-1 level was negatively correlated to the Gensini score ($r$ = −0.291, $P < 0.05$). TyG (OR = 2.687) and TyG-LAP (OR = 4.512) were the factors determining the frequency of coronary artery disease in H-type hypertension, according to multivariate logistic regression analysis.

**Conclusion:** In conclusion, the LAP and TyG indexes were observed to be closely related to the degree of CAD in H-type individuals with hypertension, which can

Corresponding author
Bin Zhang, 13792879952@163.com

better understand the pathogenesis of coronary artery disease in patients with H-type hypertension and is of great significance for guiding clinical doctors to carry out personalized treatment and management.

## INTRODUCTION

H-type hypertension, one of the essential hypertension types, is distinguished by hypertension associated with excessive homocysteine (Hcy) (≥15 µmol/L) (*Cheng, Cheng & Wei, 2020*). Investigating H-type hypertension is crucial because such individuals constitute more than 80% of all hypertension patients. The organic obstruction causes coronary atherosclerotic heart disease and the patient's coronary artery stenosis, leading to myocardial ischemia and hypoxia. The progression of the disease can lead to myocardial necrosis, with a low cure rate and high mortality rate. It can easily cause various complications, seriously threatening patients' health and life safety (*Ni et al., 2017*). According to several studies, a high level of Hcy is an independent risk factor for post-hypertensive coronary artery disease (CAD) and raises the risk of cardiovascular and cerebrovascular disease. Individuals with H-type hypertension have a five times higher risk of cardiovascular and cerebrovascular events than patients with other types of hypertension and a risk 10 times higher than that of the general population (*Cioni et al., 2016*). Hypertension, Hcy, and other cardiovascular risk factors exacerbate arteriosclerosis. As a result, early detection and intervention of hypertension in conjunction with hyperhomocysteinemia risk factors and the search for readily available and feasible clinical detection signs are critical for the prevention, treatment, and prognosis of coronary heart disease. The clinical diagnosis of CHD still takes coronary angiography as the gold standard. Although this method can simultaneously examine the degree of coronary artery lesions and vessel stenosis, its clinical implementation may be hindered due to the invasive nature of the examination and its high cost (*Karagiannidis et al., 2022*). In recent years, with the advancement of modern medical technology, the quest for biomarkers associated with disease progression has become an area of intense research. In addition to conventional obesity evaluation variables, including waist circumference (WC) and body mass index (BMI), the lipid accumulation index (LAP) is a combination of triglyceride (TG) and waist circumference (WC) that can more accurately detect the level of visceral fat accumulation (*Xu et al., 2020*). The LAP index is simple to calculate and has recently gained scholarly attention. The measurement of LAP is convenient, and many studies have proved that LAP is closely related to many chronic diseases such as diabetes, hypertension and stroke. The LAP calculation formula of these studies is derived from Kahn's research, and the formula is based on the data of the National Health and Nutrition Survey. Most of the existing studies in China use the formula proposed by Kahn, but due to differences in race, population and region, the formula is not necessarily applicable to the Chinese

population (*Wang et al., 2018*). Previous research suggests that insulin resistance (IR), which can be challenging and aggravating to diagnose, is directly related to hypertension. According to studies, the triglyceride-glucose (TyG) index can be used as a substitute for the IR index and correlates with cardiovascular disease (*Raimi et al., 2021*). Investigations on potential variables associated with CAD in patient with H-type hypertension are scarce. In order to offer theoretical and practical recommendations for preventing and treating coronary artery disease among individuals with H-type hypertension, this study examined the association between these markers and CAD in patients with H-type hypertension.

# MATERIALS AND METHODS

## Materials

The research subjects were selected from 186 individuals suffering from essential hypertension admitted to our facility between June 2020 and January 2021. All samples obtained in this study were approved by the ethics committee of the Qingdao Fuwai Cardiovascular Hospital and abided by the ethical guidelines of the Declaration of Helsinki, and ethics committee agreed to waive informed consent. The criteria for patient selection were as follows: (1) all enrolled subjects met the diagnostic criteria for essential hypertension and underwent coronary angiography, and (2) age ≥18 years. The individuals that were excluded were those (1) with missing clinical data, (2) with severe malnutrition and tumors, and/or (3) who have recently taken drugs that have an impact on Hcy metabolism.

## Methods

### Data collection

The data of individuals were acquired through the hospital information management system, including demography data such as gender, age, height, weight, blood pressure, WC, and disease-related data such as the course of the disease and medical history.

### Laboratory indexes

(1) After 8 to 12 h of fasting, blood samples were taken, and the Hitachi 747 auto-analyzer (Hitachi, Tokyo, Japan) was used to determine the following: fasting plasma glucose (FPG), high-density lipoprotein cholesterol (HDL-C), TG, fasting total cholesterol (TC), plasma homocysteine (Hcy), uric acid, and creatinine. The following formula was used to determine TyG: TyG index = ln [TG (mg/dl) × FPG (mg/dl)/2]. Male LAP is equal to [(WC-65) × TG], and female LAP is equivalent to [(WC-58) × TG]. Individuals with plasma Hcy ≥ 15 mol/L were classified as having H-type hypertension, while those with serum Hcyv < 15 mol/L were categorized as having non-H-type hypertension. (2) Glucagon like peptide-1 (GLP-1) and serum amyloid A (SAA) levels. After enrollment, peripheral elbow venous blood was collected from the subjects under fasting conditions, and serum was separated by centrifugation. Serum GLP-1 and SAA levels were measured by double antibody sandwich method. The kit was purchased from Shanghai Xinyu Bioengineering Co., Ltd (Shanghai, China).

### Coronary artery disease

(1) The traditional Judkins approach was employed to perform the selective right and left coronary angiography on a GE digital angiography system. Two experienced chief cardiac catheter physicians jointly examined the angiography results. Coronary heart disease can be characterized as a 50% narrowing of the primary coronary artery (right coronary artery (RCA), left circumflex branch, left anterior descending branch, and left main artery) and its primary branches (such as posterior descending branch, obtuse marginal branch, and diagonal branch). According to the count of the involved main coronary artery branches with stenosis ≥50%, the disease can be divided into triple vessel lesions, double vessel lesions, and single vessel lesions. Single vessel lesion: Stenosis of the left circumflex branch, left anterior descending branch, or one vessel in RCA; Double vessel lesion: Stenosis involving the left main artery, regardless of whether there is stenosis in the left circumflex branch or left anterior descending branch, are recorded as double vessel lesions; Triple vessel lesion: Stenosis connecting the left anterior descending branch, left circumflex branch, and RCA simultaneously, or left primary artery lesions accompanied by RCA lesions. (2) The Gensini scoring standard (*Yang, Liu & Xiang, 2011*) was based on the diameter of coronary artery stenosis, with 32 points for 100%, 16 points for 91% to 99%, eight points for 76% to 90%, four points for 51% to 75%, two points for 26% to 50%, and one point for 1% to 25%. There were corresponding coefficients for the lesion site: The lesion coefficient for the left main artery lesion was 5.0 points. The lesion coefficients for the anterior descending branch's distal, middle, and proximal segments were 1.0, 1.5, and 2.5, respectively. The lesion coefficients for the first and second diagonal branches was 1.0 and 1.5. The lesion coefficients for the circumflex branch's distal, middle, and proximal segments were 1.0 and 2.5, respectively. The lesion coefficients for the obtuse marginal branch were 1.0; The lesion coefficients for the distal, middle, and proximal segments of RCA and the posterior descending branch were all 1.0. The score for each lesion was equal to the sum of the stenosis degree score and the lesion site coefficient, and the patient's Gensini score was the sum of the lesion scores. Notably, the most severe stenosis was calculated for those with multiple stenoses of the same vessel. (3) Coronary artery stenosis is classified into four categories based on the amount of lumen area that has been reduced: Grade I lesion, lumen area reduction ≤25%; Grade II lesion, lumen area reduction >25–50%; Grade III lesion, lumen area reduction >50–75%; Grade IV lesion, lumen area reduction >75–100%.

## Statistical analysis

For the analysis, SPSS21.0 statistical software was employed. The measurement data in the study results were represented by mean ± SD, and the comparisons between both groups were carried out using the paired t-test. The number of cases or rates described in the counting data and their comparisons between the two groups were conducted by $\chi^2$ test. The Kruskal Wallis rank sum method was utilized to compare various groups of ranked data. Data from the univariate analysis that was statistically significant was incorporated

**Table 1 Comparison of general data between the two groups.**

| Group | n | Age (years) | Gender (Male) | BMI (kg/m$^2$) | Smoking | Systolic blood pressure (mm Hg) |
|---|---|---|---|---|---|---|
| Non-H-type hypertension group | 73 | 63.38 ± 13.46 | 52 | 22.58 ± 3.67 | 53 | 163.22 ± 5.28 |
| H-type hypertension group | 113 | 64.21 ± 12.81 | 73 | 23.25 ± 3.87 | 68 | 162.54 ± 5.12 |
| $\chi^2$/t value | | 0.423 | 0.885 | 1.176 | 3.012 | 0.874 |
| P value | | 0.673 | 0.347 | 0.241 | 0.083 | 0.383 |

| Group | n | Diastolic blood pressure (mm Hg) | FPG (mmol/L) | TC (mmol/L) | HDL-C (mmol/L) | TG (mmol/L) | Hcy (mmol/L) | GLP-1 (pmol/L) | SAA (mg/L) |
|---|---|---|---|---|---|---|---|---|---|
| Non-H-type hypertension group | 73 | 94.12 ± 6.09 | 7.24 ± 0.54 | 4.59 ± 0.38 | 1.48 ± 0.16 | 1.87 ± 0.18 | 7.98 ± 0.81 | 35.12 ± 5.98 | 8.18 ± 1.98 |
| H-type hypertension group | 113 | 95.27 ± 6.17 | 7.11 ± 0.42 | 4.47 ± 0.47 | 1.49 ± 0.15 | 1.89 ± 0.21 | 17.32 ± 1.13 | 22.65 ± 4.48 | 13.54 ± 2.76 |
| $\chi^2$/t value | | 1.248 | 1.840 | 1.829 | 0.432 | 0.670 | 61.170 | 16.221 | 14.369 |
| P value | | 0.214 | 0.067 | 0.069 | 0.666 | 0.504 | <0.001 | <0.001 | <0.00 |

into the multivariate analysis. The logistic regression model was utilized for multivariate analysis, Pearson correlation test was used for correlation analysis, and a $P < 0.05$ was regarded as a statistically significant threshold.

# RESULTS

## Comparison of the general features of both groups

There were no significant differences in diastolic blood pressure, systolic blood pressure, sex, age, BMI, smoking, FPG, TC, HDL-C, and TG between both groups ($P > 0.05$). Table 1 demonstrates a statistically significant difference ($P < 0.05$) in Hcy, GLP-1and SAA levels in both groups.

## Comparison of LAP and TyG indexes between two groups

According to Table 2, the LAP and TyG indices variations in both groups were substantially significant ($P < 0.05$).

## Comparison of the degree of CAD in both groups

Table 3 shows that the differences in the number of lesion branches, degree of stenosis, and Gensini score in both groups were considerably significant ($P < 0.05$).

## Comparison of LAP and TyG indexes in H-type hypertension patients with different lesion counts

Table 4 indicates that the LAP and TyG indexes were substantially higher ($P < 0.05$) for individuals with triple and/or double vascular lesions than those without or with single vessel lesions.

**Table 2 Comparison of LAP and TyG indexes between two groups.**

| Group | $n$ | LAP | TyG |
|---|---|---|---|
| Non-H-type hypertension group | 73 | 17.81 ± 6.14 | 6.12 ± 0.37 |
| H-type hypertension group | 113 | 26.43 ± 7.76 | 6.51 ± 0.49 |
| t value | | 8.007 | 5.812 |
| $P$ value | | <0.001 | <0.001 |

**Table 3 A comparison of the severity of coronary artery disease in both groups.**

| Item | Non-H-type hypertension group ($n$ = 73) | H-type hypertension group ($n$ = 113) | $\chi^2$/t value | $P$ value |
|---|---|---|---|---|
| Number of stenosed coronary vessels | | | | |
| No lesions | 2 | 9 | 35.223 | <0.001 |
| Single vessel lesion | 50 | 54 | | |
| Double branch lesion | 17 | 5 | | |
| Triple vessel lesion | 44 | 5 | | |
| Degree of stenosis | | | | |
| Grade I lesion | 2 | 14 | 66.443 | <0.001 |
| Grade II lesion | 15 | 40 | | |
| Grade III lesion | 42 | 9 | | |
| Grade IV lesion | 54 | 10 | | |
| Gensini score (points) | 23.57 ± 3.25 | 10.16 ± 2.78 | 29.045 | <0.001 |

**Table 4 Comparison of LAP and TyG indexes in H-type hypertension patients with different lesion counts.**

| Degree of stenosis | LAP | TyG |
|---|---|---|
| Without lesions and with single vessel lesions ($n$ = 52) | 25.87 ± 6.25 | 5.87 ± 0.42 |
| With double vessel lesions and with triple vessel lesions ($n$ = 61) | 28.91 ± 7.17 | 7.98 ± 0.58 |
| t value | 2.382 | 21.803 |
| $P$ value | 0.019 | <0.001 |

**Table 5 Comparison of LAP and TyG indexes in H-type hypertension patients with different degrees of stenosis.**

| Degree of stenosis | LAP | TyG |
|---|---|---|
| Grade I–II lesions ($n$ = 17) | 25.12 ± 6.18 | 5.98 ± 0.48 |
| Grade III–IV lesions ($n$ = 96) | 29.76 ± 7.95 | 7.76 ± 0.57 |
| t value | 2.284 | 12.125 |
| $P$ value | 0.024 | <0.001 |

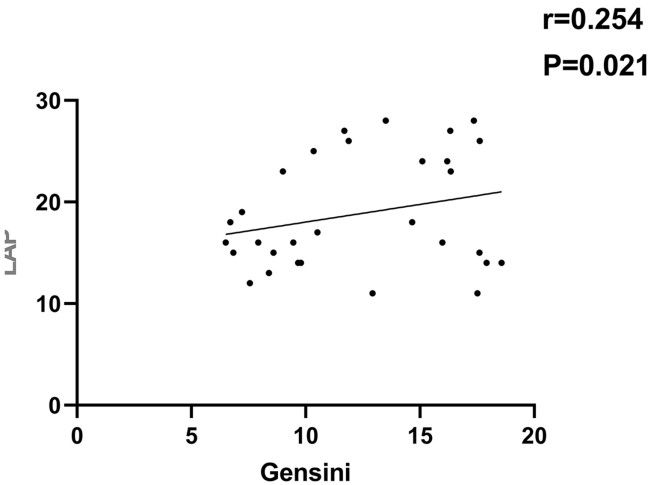

**Figure 1 Correlation analysis between LAP index and Gensini score.**

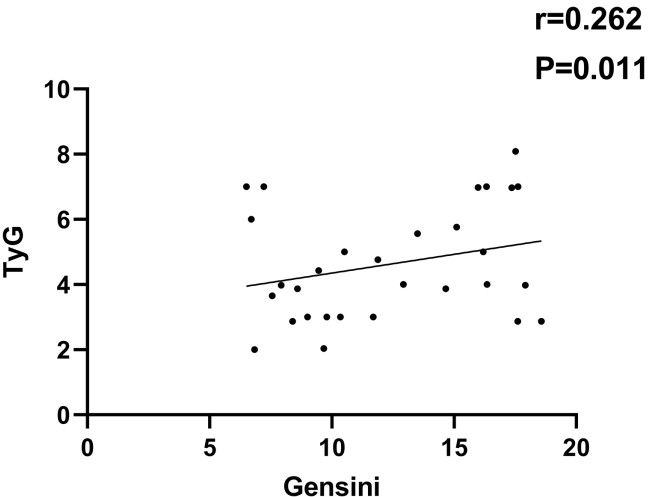

**Figure 2 Correlation analysis between TyG index and Gensini score.**

### LAP and TyG indexes were compared in H-type hypertension patients with varying degrees of stenosis

Table 5 indicates that the LAP and TyG indexes of patients with grade III-IV lesions were substantially more significant than patients with grade I–II lesions ($P < 0.05$).

### Correlation analysis

As indicated in Figs. 1–4, according to Pearson correlation analysis, the LAP, TyG indices and SAA level were adversely connected to the Gensini score ($r$ = 0.254, 0.262, 0.299, $P$ = 0.021, 0.011, 0.021), the GLP-1 level was negatively correlated to the Gensini score ($r$ = −0.291, $P$ = 0.001).

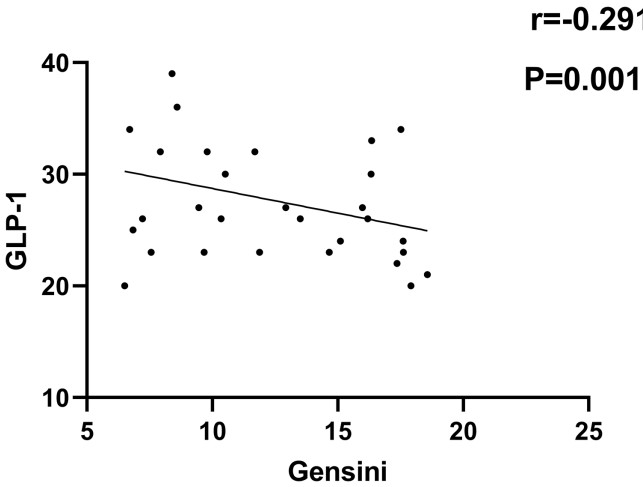

**Figure 3 Correlation analysis between GLP-1 and Gensini score.**

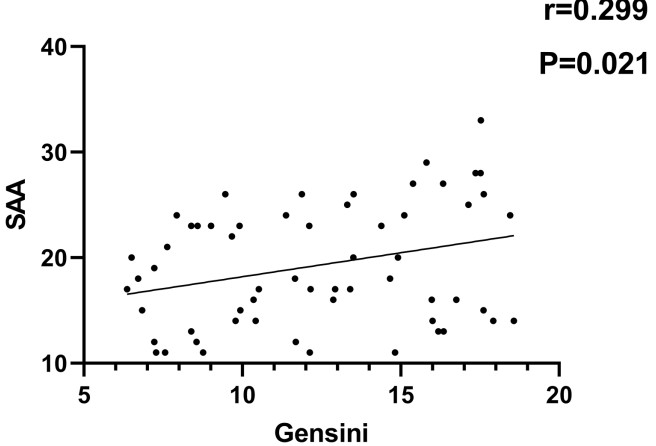

**Figure 4 Correlation analysis of SAA and Gensini score.**

**Table 6 Patients with H-type hypertension who have their risk factors for coronary artery disease.**

| Risk variable | OR | Ward value | B value | SE value | 95% CI | *P* value |
|---|---|---|---|---|---|---|
| TyG (actual value substituted) | 2.687 | 5.764 | 1.024 | 0.465 | [1.298–6.431] | <0.001 |
| LAP (actual value substituted) | 4.512 | 7.789 | 1.652 | 0.537 | [1.514–13.326] | <0.001 |

## Multiple linear regression analysis of the influencing factors of coronary artery disease in H-type hypertension patients

A multivariate logistic regression analysis was performed using statistically significant parameters from the univariate test as independent factors and stenosis grade as dependent variables (grade I–II lesions = 0, grade III–IV lesions = 1). As indicated in Table 6, the results showed that TyG (OR = 2.687) and TyG-LAP (OR = 4.512) were the variables that influenced the development of coronary artery disease in H-type hypertension ($P < 0.05$).

## DISCUSSION

Atherosclerosis is presented as the main pathological change in coronary artery disease. According to research, aberrant blood lipids (TG, TC, and LDL-C) are independent variables contributing to the disease's progression, with LDL-C changes being the most significant. The main reason is that LDL-C is easily oxidized, damaging endothelial cells' function and enhancing vascular endothelium's permeability. Furthermore, endothelial cells release plasminogen activator inhibitors, accelerating the formation of fibrous plaques (*Yan et al., 2021*; *Yeh et al., 2017*). According to the current investigation findings, individuals with H-type hypertension had a greater prevalence and severity of CAD than patients with non-H-type hypertension. This is because an increase in Hcy content will encourage the growth of vascular smooth muscle cells and impact their ability to function. These changes will also affect lipid metabolism, platelet adhesion, and the development of atherosclerosis (*Liu et al., 2022*; *Huang et al., 2017*). This is consistent with the research indicating that Hcy content is closely linked to the progression of CAD.

In the present research, individuals with H-type hypertension exhibited higher LAP and TyG indices levels than those with non-H-type hypertension. A multivariate regression analysis revealed a positive relationship between LAP, TyG indices, and CAD severity. Currently, there are many indicators for evaluating cardiovascular diseases in clinical practice, such as BMI and WC. Among these indicators, BMI mainly reflects the degree of overweight in the human body. Still, its evaluation for lean fat content is not good enough, which cannot fully reflect the individual fat distribution. patient with normal BMIs with excessive body fat may also be at higher risk for cardiovascular disease. WC is considered an accurate indicator for evaluating abdominal obesity, which is easy to obtain and has high repeatability. However, it has shortcomings distinguishing subcutaneous fat from visceral fat and is easily influenced by height (*Ebrahimi et al., 2023*; *Khanmohammadi et al., 2022*). Scholars have proposed the concept of LAP based on WC and TG and believe that it can better identify the risk of cardiovascular disease compared to BMI and WC. Insulin resistance (IR) can promote atherosclerosis and plaque progression through various mechanisms, including endothelial cell damage and down-regulation of the insulin signaling pathway (*Fukuchi et al., 2002*). Nitric oxide production and release in vascular endothelial cells are decreased in the IR state, which lowers the effect of insulin on blood flow stimulation, results in vascular endothelial cell dysfunction, and may raise the risk of atherosclerosis (*Li et al., 2022*). In addition, the standard insulin signal of vascular endothelium can prevent atherosclerosis. Vascular cell adhesion molecule-1 and intracellular adhesion molecule-1 are upregulated due to endothelial insulin receptor-AKT pathway downregulation in IR, which may be associated with the development of arterial plaque (*Alizargar et al., 2020*). Numerous clinical studies have confirmed the function of IR in coronary artery disease by demonstrating that IR has been connected with coronary artery disease (*Alizargar et al., 2020*). As one of the most prevalent geriatric diseases, hypertension may promote atherosclerosis onset and progression and is a primary risk factor for hypertensive target organ damage, including cerebral infarction, coronary heart disease, and hypertensive renal function damage. The decrease in LAP and TyG indexes

indicates hypertensive target organ damage as a risk factor (*Ramdas Nayak et al., 2022*; *Xu et al., 2022*; *Shi et al., 2021*). TyG index has been proved to be a reliable alternative indicator of IR and plays a significant role in the progression of coronary atherosclerosis. An international multi-center prospective cohort study (1,143 cases) found that during an average 4-year follow-up period, the arterial plaque burden was higher in the high TyG index group, and the higher TyG index was independently associated with the risk of coronary plaque progression (OR = 1.409, 95% CI [1.062–1.869], $P = 0.017$) and the risk of rapid coronary plaque progression (OR = 1.557,95% CI [1.109–2.185], $P < 0.011$) (*Won et al., 2020*).

## CONCLUSIONS

In patients with H-type hypertension, the LAP and TyG indexes are closely linked with the degree of CAD, which can help to better understand the pathogenesis of coronary artery disease in patients with H-type hypertension and is of great significance for guiding clinical doctors to carry out personalized treatment and management. However, there are limitations to this investigation, including a single center and a small sample size. Hence, larger sample randomized controlled studies are necessary to validate the preliminary results and to provide more convincing theoretical support for existing conclusions.

### Funding
The authors received no funding for this work.

### Competing Interests
The authors declare that they have no competing interests.

### Author Contributions
- Wenwen Yuan conceived and designed the experiments, performed the experiments, prepared figures and/or tables, authored or reviewed drafts of the article, and approved the final draft.
- Yan Shao analyzed the data, prepared figures and/or tables, and approved the final draft.
- Dong Zhao conceived and designed the experiments, performed the experiments, analyzed the data, prepared figures and/or tables, authored or reviewed drafts of the article, and approved the final draft.
- Bin Zhang performed the experiments, analyzed the data, authored or reviewed drafts of the article, and approved the final draft.

### Human Ethics
The following information was supplied relating to ethical approvals (*i.e.*, approving body and any reference numbers):

All samples obtained in this study were approved by the ethics committee of the Qingdao Fuwai Cardiovascular Hospital and abided by the ethical guidelines of the Declaration of Helsinki.
## Data Availability

The raw data are available in the Supplemental File.

## Supplemental Information

Supplemental information for this article can be found online at http://dx.doi.org/10.7717/peerj.16069#supplemental-information.

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
