# Peer review of "Correlation analysis of lipid accumulation index, triglyceride-glucose index and H-type hypertension and coronary artery disease"

_PeerJ, doi:10.7717/peerj.16069_

## Round 0.1 · original submission · Minor Revisions

Here are my comments, and please revise your manuscript according to the comments of our Reviewers and my suggestions:

1. Line 46: The unit "mol/L" or "μmol/L". Please carefully confirm this.
2. Replace people with patient, when necessary.
3. Show the detailed p-values and r values in Figure 1, 2, 3 and 4.
4. The Abstract as well as the Conclusion section doesn't explicitly outline the significance of this study. What are the implications and benefits of the findings of this study for a better understanding of the mechanisms of the disease as well as for improving the associated treatments for clinicians?
5. Revise "individuals comprise more than 80% of all hypertension patients" to "individuals constitute more than 80% of all hypertension patients" for improved clarity.
6. Revise "and a 10 times higher risk than the general population" to "and a risk 10 times higher than that of the general population" for proper comparison structure.
7. Revise "its clinical implementation will be hampered due to its invasive examination and high price" to "its clinical implementation may be hindered due to the invasive nature of the examination and its high cost" for better phrasing.
8. Revise "that insulin resistance (IR), however challenging and aggravating to diagnose" to "that insulin resistance (IR), which can be challenging and aggravating to diagnose" for improved clarity.

Reviewer 1 ·

Excellent Review

This review has been rated excellent by staff (in the top 15% of reviews)
EDITOR COMMENT
Dear Reviewer, Thank you for taking the time to review this manuscript. Your careful and comprehensive assessment is highly appreciated. You have noticed several strengths of the manuscript, such as the clear presentation of the research population, suitable statistical analysis, and intriguing results. Your constructive and objective comments and suggestions will greatly enhance the impact and scientific rigor of this manuscript. We acknowledge your recommendation for addressing the mentioned comments and suggestions, as well as providing more extensive discussions and practical implications to strengthen the manuscript and provide valuable insights. We are grateful for your suggestion to re-analyze the data and introduce relevant studies and literature to further improve the paper. Your input will undoubtedly contribute to its overall quality and relevance. Once again, we would like to express our gratitude for your effort and valuable contributions. Your review will not only help improve this particular manuscript but will also motivate future reviewers in their evaluation of scientific research. Thank you for your dedication to the peer-review process. Best regards, Handling Editor

Basic reporting

1.1 The manuscript lacks a clear and concise introduction that outlines the purpose, significance, and scope of the study. Providing a context to the readers will help them understand the relevance of the findings.
1.2 The research questions or hypotheses should be explicitly stated, guiding the reader through the study objectives.
1.3 The manuscript requires proofreading for grammatical errors, typographical mistakes, and sentence structure to improve overall readability.
1.4 Detailed information on the sample size, inclusion and exclusion criteria, and demographics of the study population should be provided to allow for better reproducibility and generalization of the findings.

Experimental design

2.1 The study lacks a control group of non-hypertensive individuals, which would have assisted in identifying the specific effects of H-type hypertension on the LAP and TyG indexes.
2.2 Why did the comparisons between both groups were carried out using the paired t-test? How to make the “pair”?
2.3 Describe how missing data was handled, if any, and provide a rationale for the chosen approach, such as imputation or exclusion.
2.4 It would be beneficial to outline any potential sources of bias or limitations in the study design, such as selection bias or confounding variables.

Validity of the findings

3.1 While the reported p-values suggest statistical significance, effect sizes or confidence intervals should be provided to demonstrate the magnitude and precision of the relationships.
3.2 The results of the multiple linear regression analysis described in Section 3.7 would be strengthened by reporting the goodness of fit statistics (e.g., R-squared).
3.3 Address the potential limitations of using LAP and TyG indexes as predictors of coronary artery disease. Discuss the generalizability of these findings to a larger population.

Additional comments

4.1 Provide a more comprehensive discussion of previous literature in the field to give readers a better understanding of the current study's contributions and novelty.
4.2 Elaborate on the clinical implications and how the LAP and TyG indexes can be practically utilized in the diagnosis, prognosis, or management of patients with H-type hypertension.
4.3 Discuss the potential mechanisms underlying the observed associations between LAP, TyG indexes, and coronary artery disease. This will help contextualize the findings and promote a deeper understanding of the underlying biology.
4.4 Clarify the practical implications of the LAP and TyG indexes and how they might be integrated into clinical practice. Are these indexes suitable for routine screening or monitoring of coronary artery disease in H-type hypertension patients?

Reviewer 2 ·

Basic reporting

The manuscript provides a comprehensive introduction and background, clearly explaining how the study fits into the broader field of knowledge. The authors effectively establish the relevance of their research by highlighting the relationship between aberrant blood lipids, hypertension, and the development of coronary artery disease (CAD). They correctly identify the knowledge gap regarding the specific evaluation of LAP and TyG indexes in H-Type hypertension patients with CAD, indicating the need to fill this gap with their study.

Experimental design

1) Please explain why age ≥ 18 years was considered as a criterion for patient selection. Specify any particular significance or rationale.
2) Author could consider mentioning any efforts made to address potential biases in subject selection.
3) Please mention how to ensure the accuracy and reliability of the diagnostic criteria used for essential hypertension.

Validity of the findings

1) Clarify the units of measurement for LAP and TyG indexes in Tables 2, 4, 5, and 6.
2) Consider adding additional statistical measures, such as confidence intervals, in Tables 2, 4, 5, and 6 to provide a more comprehensive interpretation of the data.
3) Consider suggesting future research directions based on the limitations or unanswered questions raised by the study.
4) Overall, the technical standard employed in the study is commendable, providing sufficient detail for replication. The methodology is well-structured, and the statistical analysis is appropriate, utilizing relevant tests such as t-tests, chi-square tests, and correlation analysis. However, additional information on sample size calculation, power analysis, and justification would improve the manuscript's replicability.

Additional comments

The manuscript demonstrates clear and technically correct English expression throughout. The introduction and background provide a comprehensive overview of the study's context within the broader field of knowledge. The authors effectively identify the knowledge gap by highlighting the limitations of current indicators used to evaluate cardiovascular diseases. They explain how their study fills this gap by introducing the LAP and TyG indices as potential improved predictors of cardiovascular risk. The technical standard employed in the study appears to be of high quality, with well-defined methods and statistical analyses. The level of detail provided is sufficient to allow for potential replication of the study. The availability and robustness of the underlying data are not clearly discussed in the manuscript, and it would be beneficial for the authors to provide more information on data sources and collection methods. Overall, the manuscript demonstrates a strong foundation and contributes to advancing knowledge in the field.

Reviewer 3 ·

Basic reporting

The authors appropriately establish the connection between aberrant blood lipids, hypertension, and the progression of coronary artery disease (CAD). The knowledge gap regarding the specific association between LAP and TyG indexes and CAD severity in H-Type hypertension patients is clearly identified, demonstrating the need for this study.

Experimental design

I. Provide a brief description of the sample size calculation or power analysis to justify the chosen participant numbers for each group.
II. Specify how the selected individuals were enrolled in the study (e.g., consecutive sampling, random sampling) to provide transparency and address potential sampling bias.
III. Did stratified sampling or random sampling were used to provide transparency and address potential sampling bias?
IV. Provide a justification or rationale for the choice of the cutoff values used in the categorization process.

Validity of the findings

I. Provide a brief discussion and interpretation of the results in the text, highlighting the clinical significance of the observed differences.
II. Include a reference range or normal values for LAP and TyG indexes, if available, to facilitate interpretation and comparison with other studies.
III. Consider presenting additional descriptive statistics, such as median and interquartile range, for variables that may not follow a normal distribution.
IV. Provide a more detailed discussion of the limitations of the study, including potential sources of bias or confounding factors.

Additional comments

I. Consolidate and streamline the language used throughout the manuscript to improve readability and comprehension.
II. Provide additional information on the control measures employed to mitigate potential confounding factors.
III. Consider expanding on the limitations section.
IV. Address any inconsistencies or ambiguities in terminology to improve clarity and precision.
V. Clarify the units of measurement for all variables reported in the manuscript.

---

## Round 0.2 · accepted · Accept

In carefully evaluating the content of this revised paper, I was satisfied with the responses and revisions made by the authors. The Reviewer's concerns have been well addressed. With the necessary revisions and improvements, the quality of this paper has been significantly improved. I believe that this revised manuscript is ready to be considered for publication in this journal.